# CHRONOLOGICAL THINKING IN FULL-DUPLEX SPOKEN DIALOGUE LANGUAGE MODELS

## ABSTRACT

Recent advances in spoken dialogue language models (SDLMs) reflect growing interest in shifting from turn-based to full-duplex systems, where the models continuously perceive user speech streams while generating responses. This simultaneous listening and speaking design enables real-time interaction and the agent can handle dynamic conversational behaviors like user barge-in. However, during the listening phase, existing systems keep the agent idle by repeatedly predicting the silence token, which departs from human behavior: we usually engage in lightweight thinking during conversation rather than remaining absent-minded. Inspired by this, we propose *Chronological Thinking*, a on-the-fly conversational thinking mechanism that aims to improve response quality in full-duplex SDLMs. Specifically, chronological thinking presents a paradigm shift from conventional LLM thinking approaches, such as Chain-of-Thought, purpose-built for streaming acoustic input. (1) *Strictly causal*: the agent reasons incrementally while listening, updating internal hypotheses only from past audio with no lookahead. (2) *No additional latency*: reasoning is amortized during the listening window; once the user stops speaking, the agent halts thinking and begins speaking without further delay. Experiments demonstrate the effectiveness of chronological thinking through both objective metrics and human evaluations show consistent improvements in response quality. Furthermore, chronological thinking robustly handles conversational dynamics and attains competitive performance on full-duplex interaction metrics.

## 1 INTRODUCTION

Speech is a natural and fundamental modality for human–computer interaction, offering intuitive, efficient, and expressive communication (Cui et al., 2024; Huang et al., 2025). Reflecting this importance, spoken dialogue language models (SDLMs) have become increasingly central in AI as advanced systems seek to support natural interaction. In academia, SDLMs remain an active area of research (Nguyen et al., 2023; Hu et al., 2025; Ding et al., 2025), with a growing emphasis on end-to-end speech-to-speech dialogue systems that integrate speech understanding and generation within a unified interactive loop.

More recently, full-duplex models have garnered significant attention as a novel SDLM architecture (Défossez et al., 2024; Chen et al., 2025b;a), departing from traditional turn-based interaction by removing rigid listen-then-speak alternation (Veluri et al., 2024a; Lin et al., 2022; Liao et al., 2025), as illustrated in Figure 1. In a full-duplex system, the model continually ingests streaming user speech while synthesizing the corresponding response in real time. This "always-on" agent delivers more natural, fluid, and human-like conversations, with the ability to proactively take turns, offer backchannel responses, make timely corrections, and gracefully yield with user barges-in (Défossez et al., 2024).

However, despite active exploration of full-duplex models, we identify a common issue across existing designs: the agent is kept idle during user speaking by repeatedly predicting a "silence token". This practice is problematic for two reasons: (1) in autoregressive models, prolonged repetition of a single token can be harmful, biasing the next-token distribution and reinforcing degeneracy (Zhu et al., 2023; Guan & Huang, 2023; Xu et al., 2022); and (2) it leaves the listening window underutilized, missing opportunities to form intent hypotheses and organize the forthcoming response. In

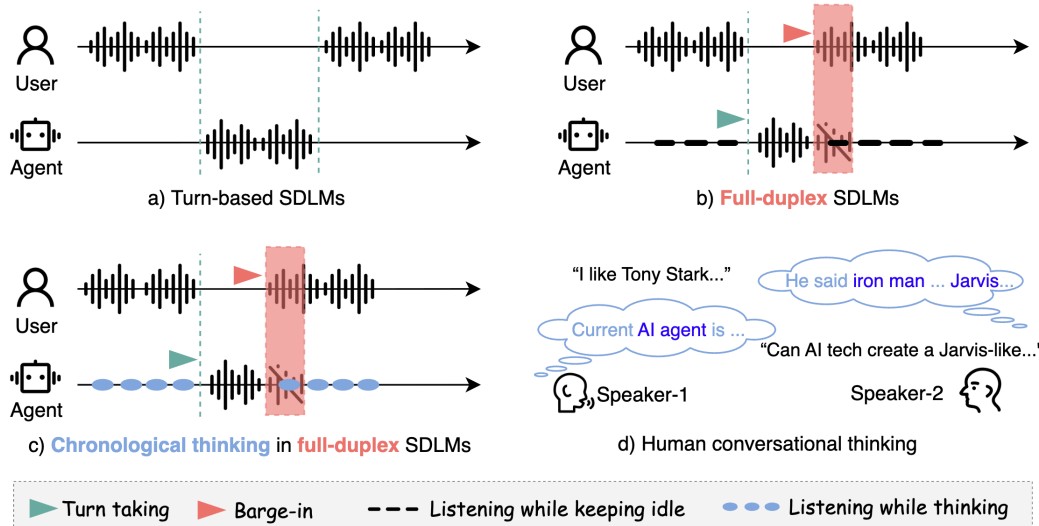

Figure 1: Comparison of a) turn-based SDLMs, b) full-duplex SDLMs c) chronological thinking in full-duplex SDLMs (ours) and d) human conversational thinking patterns.

contrast, human listeners perform lightweight, conversational thinking while listening—continually updating beliefs about the speaker's intent and sketching response structure, as shown in Figure 1. This observation raises a central research question: *Can such on-the-fly thinking be feasible within full-duplex SDLMs?*

Notwithstanding the above, implementing such a mechanism under full-duplex constraints is particularly challenging. First, streaming user speech imposes strict causality: the agent must reason incrementally while listening, updating hypotheses only from past audio without lookahead or access to a complete utterance (Veluri et al., 2024a). Second, since user speech can end at any moment, the reasoning process must be preemptible and amortized during listening; once the user stops, the agent should transition to speaking immediately without incurring additional latency (Chiba & Higashinaka, 2025). However, existing "thinking" techniques in LLMs, such as Chain-of-Thought (CoT) (Wei et al., 2022), are typically lengthy and post hoc, and therefore not directly applicable to the full-duplex setting. These limitations motivates the development of a new paradigm.

To address these challenges, we propose CT-SDLM, a full-duplex **SDLM** with a **C**hronological **T**hinking mechanism. Inspired by the Adaptive Control of Thought-Rational (ACT-R) theoretical framework, we propose different node types corresponding to specific modules in the typical ACT-R architecture (Ritter et al., 2019), which replace the repeated silence tokens in conventional full-duplex system. Given the streaming user speech input, these nodes are chronologically predicted based on real-time semantic segments, ensuring causality in the thinking process. This design is consistent with human conversational behavior, where we tend to form associations and generate responses incrementally based on the semantic fragments of the interlocutor's speech, as shown in Figure 1(d). Furthermore, compared with typical long and ad hoc language chain, the structured node representation significantly reduces the token cost of reasoning while retaining useful information in auto-regressive generation, even if the user stops speaking abruptly. Therefore, this compact thinking chain is preemptible, allowing the system to seamlessly switch to response generation without incurring any additional latency. We conduct both objective and subjective evaluations to verify the effectiveness of chronological thinking in full-duplex SDLMs. Across task-oriented dialogue (Si et al., 2023; Yan et al., 2025) and open-domain spoken QA (Nachmani et al., 2023; Berant et al., 2013) benchmarks, CT-SDLM consistently outperforms strong baselines in both A/B tests and quantitative metrics. In addition, evaluations on full-duplex interaction metrics confirm that CT-SDLM introduces no additional latency in turn taking and user barge-in, demonstrating its robustness to conversational dynamics.

Our contributions are summarized as follows: (1) We propose a full-duplex SDLM with chronological thinking—a strictly causal, on-the-fly reasoning mechanism that enables the model to incrementally process semantic segments during user speech. (2) Our design yields a compact, preemptible

thinking process that replaces redundant silence tokens without adding latency, and achieves consistent gains over baselines in both subjective and objective evaluations. (3) We demonstrate that chronological thinking serves as a viable new paradigm for full-duplex interaction, with the potential to influence future directions in real-time dialogue modeling and human-machine communication.

## 2 RELATED WORK

**Full-Duplex Spoken Dialogue Systems.** Early spoken dialogue systems involved turn-based architectures (Sarikaya et al., 2002), where user speech input and system output occurred sequentially. Recent advances have shifted toward full-duplex dialogue systems (Veluri et al., 2024b; Wang et al., 2024; Ma et al., 2025), enabling the agent to simultaneously listen and speak. Research in this area has largely focused on engineering challenges such as streaming ASR (Moritz et al., 2020; He et al., 2019; Yamamoto et al., 2025), incremental TTS (Chiba & Higashinaka, 2025; Skerry-Ryan et al., 2018), and mechanisms for barge-in handling (Chen et al., a; Schlangen & Skantze, 2011). In addition, studies on incremental dialogue management (Khouzaimi et al., 2016; Zhang et al., 2025) have explored how conversational agents can respond more naturally in overlapping speech conditions. Nevertheless, during the listening phase, existing systems primarily enforce silence by repeatedly predicting pause or silence tokens. This diverges from naturalistic human conversational behavior, where silent reasoning typically occurs.

**Reasoning in Language Models.** A parallel line of work explores how language models perform reasoning through explicit intermediate steps. CoT (Wei et al., 2022) and its extensions, such as Self-Consistency (Wang et al., 2023) and Tree-of-Thoughts (Yao et al., 2023), have shown that reasoning traces improve performance across arithmetic, logic, and commonsense reasoning. Recent methods such as Program-Aided Language Models (PAL) (Gao et al., 2023) and Toolformer (Schick et al., 2023) further highlight the benefits of externalized or structured reasoning. However, these approaches are designed for static text inputs, assuming access to the full problem before reasoning begins. They often rely on non-causal computation with hypothesis revision, which is incompatible with streaming conversational input.

**Incremental and Streaming Reasoning.** Another relevant line of work investigates reasoning and generation under streaming or incremental input (Calimeri et al., 2021). In simultaneous machine translation, prefix-to-prefix frameworks (Ma et al., 2019) and monotonic attention models (Ma et al., 2020; Arivazhagan et al., 2019) have been developed to balance accuracy and latency. Similar ideas in incremental decoding (Dalvi et al., 2018) allow models to generate partial outputs while processing incomplete inputs. In the LLM era, recent approaches such as StreamingLLM (Xiao et al., 2024), Medusa decoding (Cai et al., 2024), and attention sink methods (Xiao et al., 2024) have examined how large models can operate efficiently under bounded memory and real-time constraints. These studies illustrate the feasibility of causal reasoning under partial input but rarely consider spoken dialogue. Our work addresses this gap by introducing chronological thinking, a causal reasoning mechanism tailored for full-duplex spoken dialogue, enabling models to think continuously while listening without delaying the onset of response generation.

## 3 METHOD

In this section, we introduce the proposed full-duplex SDLM with chronological thinking mechanism. We first describe the overall network architecture, which enables simultaneous input processing and response generation, and then detail the chronological thinking mechanism that replaces redundant silence during periods when the agent listens to the user. This enhancement aims to improve the model's intelligence, enabling it to possess human-like ability of thinking while listening.

### 3.1 MODEL ARCHITECTURE

The network architecture of this paper is illustrated in Figure 2. It accepts two input streams: the user speech stream and the agent speech and text stream. The user's speech stream is input into a streaming speech encoder operating at a frame rate of $12.5$Hz, producing continuous embeddings $\boldsymbol{X} \in \mathbb{R}^{T}$, where $T$ denotes the number of frames. These embeddings are projected by a modality adapter and then summed with the embeddings of agent text tokens before being fed into an LLM

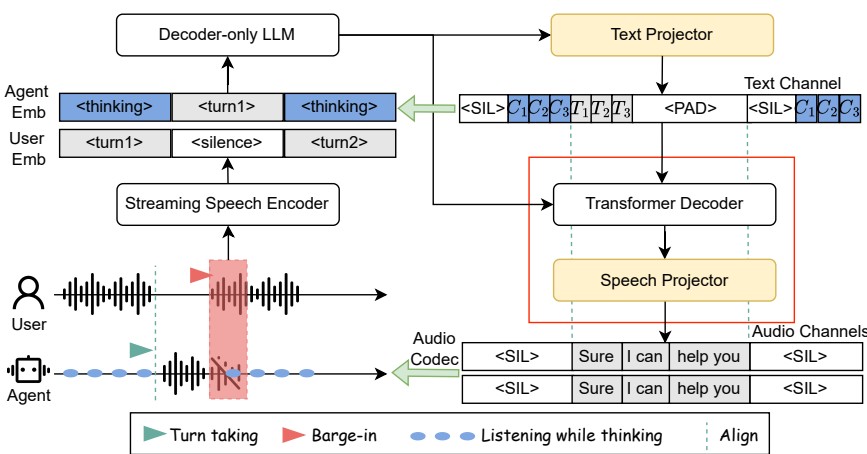

Figure 2: The network architecture of CT-Duplex with **chronological thinking** mechanism. Our model consists of a streaming speech encoder, a speech codec, an LLM backbone and a Transformer decoder. Compared to existing full-duplex SDLMs, chronological thinking is introduced during the listening phase.

backbone. To reduce the prediction burden on the LLM, instead of generating both the text and speech tokens with LLM (Hu et al., 2025), we only set the agent's text tokens $\boldsymbol{Y}^{\mathrm{txt}}$ as the prediction target for the LLM. This autoregressive process can be formulated as:

$$P_{\theta_l}(\boldsymbol{Y}^{\mathrm{txt}}|\boldsymbol{X}) = \prod_{t=1}^{T} P_{\theta_l}(Y_t^{\mathrm{txt}}|\langle \boldsymbol{Y}_{1:t-1}^{\mathrm{txt}}, \boldsymbol{X}_{1:t-1}\rangle), \tag{1}$$

where $\theta_l$ denotes the parameters of the LLM backbone. After the LLM backbone generating the agent's text tokens, an autoregressive Transformer decoder is employed to predict the agent's speech tokens $\boldsymbol{Y}^{\mathrm{spc}}$. The inputs to this Transformer decoder include the agent's text tokens $\boldsymbol{Y}^{\mathrm{txt}}$ and previously predicted agent speech tokens $\boldsymbol{Y}^{\mathrm{spc}}$, conditioned on the LLM's last hidden states $\boldsymbol{h}$, which can be formulated as:

$$P_{\theta_d}(\boldsymbol{Y}^{\mathrm{spc}}|\langle \boldsymbol{X}, \boldsymbol{Y}^{\mathrm{txt}}, \boldsymbol{h}\rangle) = \prod_{t=1}^{T} P_{\theta_d}(Y_t^{\mathrm{spc}}|\langle \boldsymbol{Y}_{1:t-1}^{\mathrm{spc}}, \boldsymbol{Y}_{1:t-1}^{\mathrm{txt}}, \boldsymbol{h}_t\rangle), \tag{2}$$

where $\theta_d$ represents the parameters of the Transformer decoder and $\boldsymbol{h}_t$ is LLM's hidden state at time step $t$. We employ Nanocodec (Casanova et al., 2025b) with Finite Scalar Quantization (Casanova et al., 2025a) to obtain agent speech tokens, generating speech codes at 12.5Hz. We jointly finetune the streaming speech encoder, LLM backbone, and autoregressive Transformer decoder using a multi-channel next token prediction training objective (Brown et al., 2020).

## 3.2 AGENT TEXT AND SPEECH ALIGNMENT

Existing methods aligns the agent's text and speech at the token level, prepends the <BOS> token at the beginning of an agent's turn, appends the <EOS> token at the end of the agent turn, and uses text padding tokens <PAD> to pad the gap between text and speech tokens (Défossez et al., 2024; Hu et al., 2025). Furthermore, during the user's turn, most full-duplex models require the LLM to predict silence tokens <SIL> to fill the agent's stream. The expected agent's text tokens in the $i$-th turn can be represented as:

$$\begin{aligned} \boldsymbol{Y}^{\mathrm{txt}} = [&< \mathtt{SIL} >, \cdots, < \mathtt{SIL} >, < \mathtt{SIL} >, \cdots, < \mathtt{SIL} >, \\ &< \mathtt{BOS} >, R_{i,1}, R_{i,2}, \cdots, R_{i,T}, < \mathtt{PAD} >, \cdots, < \mathtt{PAD} >, < \mathtt{EOS} >] \end{aligned} \tag{3}$$

where $R_{i,t}$ is the response token at time step $t$ in turn $i$. When the agent needs to listen to the user's speech, the LLM repeatedly generates silence tokens <SIL>. When the user finishes speaking and it becomes the agent's turn to respond, the LLM outputs a <BOS> token, marking the beginning of the agent's turn. When the agent's turn ends or is interrupted by the user, the LLM outputs

Table 1: The definitions of the five node types in chronological thinking chains and their corresponding relationships with different modules in the ACT-R theory

| Node Type | DESCRIPTION | ACT-R |
|---|---|---|
| Entity | Extracts entities from the dialogue. | Visual module |
| Intent | Represents the user's goal | Goal module |
| Action | Denotes the agent's executable operation | Manual module |
| Knowledge | Retrieves factual or procedural knowledge. | Declarative module |
| Logic | Captures rules or logic generated by the agent | Production system |

an `<EOS>` token, after which it resumes repeatedly generating silence tokens. This mechanism of outputting silence segments to maintain a listening state is a common paradigm in end-to-end full-duplex SDLM systems (Défossez et al., 2024; Yu et al., 2024; Hu et al., 2025).

However, forcing an LLM to repeatedly predict the same silence token not only lives the listening window unexploited but also degrades its performance by biasing the next-token distribution (Xu et al., 2022). In fact, during human conversation, when listening to others speak, the brain does not simply remain idle; instead, it engages in thinking while processing the speaker's input. Similarly, a SDLM should not continuously output silence tokens during the user's turn. Instead, it should perform the thinking process as the user's input unfolds streamingly.

### 3.3 CHRONOLOGICAL THINKING

A straightforward approach to streaming thinking is to use the streaming ASR transcript text as the thinking content (Moritz et al., 2020; He et al., 2019; Yamamoto et al., 2025). We discuss the comparison between this method and the approach proposed in this paper in Appendix A.1. Inspired by research in human cognitive architecture, particularly the ACT-R theory, which divides human cognition into distinct modules, including the visual module for recognizing entities, the goal module for maintaining current intentions, the declarative module for retrieving knowledge, the manual module for controlling actions, and the production system for managing production rules (Ritter et al., 2019), we define five distinct types of nodes, which form a chain-structured chronological thinking content. The node types and their corresponding relationships with different modules in the ACT-R framework are shown in Table 1.

The chronological thinking chain grows with the user's input. Each time a semantic segment from the user is received, the agent obtains one or more of the five types of nodes. These nodes could be any one of these five node types. There is no fixed order for the nodes. **The type of node generated depends solely on the semantics**. For example, if the user input is "Help me order a restaurant to celebrate my birthday this weekend.", a chronological thinking chain would be:

```
[Help me]{INTENT} Request assistance
[order a restaurant]{ACTION} Initiate booking process
[to celebrate my birthday]{LOGIC} Purpose: birthday celebration
[this weekend]{ENTITY} Timeframe: weekend
{KNOWLEDGE} Birthday: Decorations, discounts, or special perks
```

A complete dialogue containing chronological thinking chains is shown in Appendix A.2. The transcribed speech content within [·] is only used during the data generation phase to control the generation of chronological thinking chains and is not included in the input and output stream of the full-duplex SDLM. The content within {·} represents the node type, followed by the node's attributes. For a full-duplex SDLM, a chain node is formatted as:

```
{Node type} Node attributes
```

We employ the Qwen2.5-72B-Instruct LLM model to generate chronological thinking chains based on input dialogue data (Team, 2024). After obtaining the chronological thinking chain, we convert it into tokens, denoted as $C$, prepend the starting token `<BOC>` and append the ending token `<EOC>`, then put them to the positions originally occupied by silence tokens. Considering causality and latency requirements, we adopt the following strategies of varying lengths of $C$:

We first define the length of chronological thinking chain tokens as $M$ and the original silence tokens's length as $S$. For cases where $M$ is less than or equal to $S - 2$, we replace the last $M + 2$ silence tokens with <BOC>, thinking chain tokens, and <EOC>. This ensures that the thinking tokens appear as late as possible, striving to ensure that the thinking tokens corresponding to a semantic segment appear later than the semantic segment in user's speech. Thus, the expected agent text tokens in the turn $i$ can be expressed as:

$$\boldsymbol{Y}_i^{\text{txt}} = [<\texttt{SIL}>, \cdots, <\texttt{SIL}>, <\texttt{BOC}>, C_{i,1}, C_{i,2}, \cdots, C_{i,M}, <\texttt{EOC}>, \\ <\texttt{BOS}>, R_{i,1}, R_{i,2}, \cdots, R_{i,T}, <\texttt{PAD}>, \cdots, <\texttt{PAD}>, <\texttt{EOS}>], \tag{4}$$

where $C_{i,m}$ is the $m$-th chronological thinking chain token in the turn $i$, and the number of <SIL> equals to $S - (M + 2)$.

When $M$ is greater than $S - 2$, we first tokenize each chain node into tokens, and denote the number of tokens for each node as $M_1, M_2, \cdots, M_N$, where $N$ is the number of nodes. We then retain the first $n$ nodes such that $M' = \sum_{j=1}^{n} M_j \leq S - 2$, and $M' + M_{n+1} > S - 2$. We replace the last $M' + 2$ silence tokens with chronological thinking chain tokens formed by the first $n$ nodes, as well as <BOC> and <EOC>. Thus, the expected agent text token in the $i$-th turn can be written as:

$$\boldsymbol{Y}_i^{\text{txt}} = [<\texttt{SIL}>, \cdots, <\texttt{SIL}>, <\texttt{BOC}>, C_{i,1}, C_{i,2}, \cdots, C_{i,M'}, <\texttt{EOC}> \\ <\texttt{BOS}>, R_{i,1}, R_{i,2}, \cdots, R_{i,T}, <\texttt{PAD}>, \cdots, <\texttt{PAD}>, <\texttt{EOS}>], \tag{5}$$

where the number of <SIL> equals to $S - (M' + 2)$. We discuss the completeness of thinking chains in Appendix A.3. Finally, we use the text tokens with chronological thinking chains and the speech tokens to train the SDLM with multi-channel next token prediction (Brown et al., 2020).

## 4 EXPERIMENTS

### 4.1 DATA GENERATION

Existing real-world conversational datasets, such as Fisher conversation dataset (Cieri et al., 2004), focus mainly on casual conversations, are insufficient to train the SDLM to respond to diverse human inquiries (Défossez et al., 2024; Hu et al., 2025; Chen et al., b). To enhance the model's reasoning capabilities, we generate challenging dialogue data through synthetic methods. We first use seed content and an LLM to generate textual conversations, then convert these conversations into speech using a multi-speaker TTS system with voice cloning capabilities.

To create data for general conversation, we first curate a wide range of topics from sources like Wikipedia, covering general knowledge, common sense, and current events. These topics serve as seeds for Qwen2.5-72B-Instruct LLM (Team, 2024) to generate the textual dialogues, formulating a dataset named *GenConv*.

Table 2: Hours of synthetic training data

| Dataset | Hours |
|---|---|
| GenConv | 10.5k |
| SpokenWOZ-G | 2.0k |
| Llamaq-G | 2.7k |

To train the model's ability in scenarios requiring reasoning, we introduce SpokenWOZ and select its training set as the seed dataset and create *spokenWOZ-G*. SpokenWOZ encompasses various reasoning scenarios, including those requiring cross-turn information, temporal, mathematical, and semantic reasoning (Si et al., 2023). We prompt Qwen2.5-72B-Instruct to generate topically-related dialogues with SpokenWOZ's format. We then calculate the similarity between the SpokenWOZ's dialogues and generated dialogues using the *thefuzz* python library and discard any generated dialogue with a similarity score over 90%. Similarly, to enhance the model's knowledge base, we create the *Llamaq-G* dataset by applying the same generation scheme to generate dataset with format like LLama Questions (Nachmani et al., 2023).

After generating the textual dialogues and their corresponding chronological thinking chains, we synthesize the audio using Step-Audio-TTS-3B (Huang et al., 2025), a large-scale text-to-speech model capable of high-quality voice cloning. To ensure speaker diversity, we build a speaker library by collecting and cleaning over 50,000 single-speaker audio clips from various sources such as short-form videos, streaming media, and podcasts to serve as the prompt audio. For each dialogue,

Table 3: Performance on SpokenWOZ and MtBenchEval in terms of GPT score, BLEU, and Sentence-BERT. GT-LM is an optimal cascaded system that feeds ground-truth user turns to the LLM. We use "$thk$" to denote the proposed chronological thinking. The results of SALM-Duplex are reproduced by ourselves.

| Method | SpokenWOZ | | | MtBenchEval | | |
|---|---|---|---|---|---|---|
| | GPT score | BLEU | Sentence-BERT | GPT score | BLEU | Sentence-BERT |
| GT-LM | 2.48 | 7.60 | 0.55 | 3.15 | 10.18 | 0.73 |
| SALM-Duplex* | 2.11 | 8.73 | 0.34 | 2.25 | 5.31 | 0.47 |
| CT-Duplex w/o $thk$ | 2.40 | 12.92 | 0.52 | 2.39 | 7.00 | 0.64 |
| CT-Duplex w/ $thk$ | **2.61** | **16.30** | **0.59** | **2.44** | **7.34** | **0.67** |

two distinct speakers are randomly selected from this library, and their voices are cloned using Step-Audio-3B to produce a complete, natural-sounding multi-turn spoken conversation (Huang et al., 2025). The total hours for each dataset are summarized in Table 2.

We follow the method in (Hu et al., 2025) to create barge-in events: each dialogue turn has a random 50% probability of cutting off the agent's speech to allow the user to barge in. When a barge-in occurs, a 0.64s delay is enforced before the agent ceases speaking. We introduce a delay of 0.32s between the end of the user's speech and the start of the agent's response to enhance the naturalness of the dialogue. As demonstrated in (Hu et al., 2025), this approach enables the model to effectively learn the barge-in behavior.

## 4.2 EXPERIMENTAL SETTINGS

The model is implemented using the NeMo Toolkit (Kuchaiev et al., 2019) [1] and trained on 8 L40s (48G) GPUs. The LLM backbone is initialized from the Qwen2.5-1.5B-Instruct (Team, 2024). The speech encoder, text tokenizer and speech codec follow the ones in (Hu et al., 2025). The optimizer is AdamW with an inverse Square Root Annealing learning rate schedule. The learning rate starts from 3e-4 with a warm-up of 2500 steps. We use Whisper-large-v3 to transcribe the generated speech into text for calculating evaluation metrics (Radford et al., 2022).

## 4.3 EVALUATION DATA AND METRICS

We utilize SpokenWOZ to validate the model's response quality in scenarios requiring reasoning (Si et al., 2023). Additionally, we employ the MtBenchEval from URO-Bench, a multi-turn dialogue evaluation dataset assess the model's performance in daily conversations without complex reasoning (Yan et al., 2025). We employ the GPT scores generated by `gpt-4o-mini`, ranging from 0 to 5 to evaluate the performance. The prompts used is from URO-Bench (Yan et al., 2025). We also utilize the text BLEU score and Sentence-BERT similarity to evaluate the similarity between the generated responses and the target content (Papineni et al., 2002; Reimers & Gurevych, 2019).

To evaluate the model's factual knowledge capability, we introduce the Llama Questions and Web Questions datasets (Berant et al., 2013; Nachmani et al., 2023). The metrics utilized is accuracy.

We follow (Chen et al., b) to evaluate the turn-taking and barge-in performance of the full-duplex SDLM. The metrics include: (1) Turn-taking latency: The delay in the agent's response to the user's query in the first dialogue turn; (2) Barge-in latency: The time between the user's interruption and the agent stopping speech; (3) Barge-in success rate: The percentage of cases where the agent stops speaking within 1.5s after the user interrupts; We employed the *impatient* dataset in (Chen et al., b), where interruptions occur approximately every 2 seconds on average, to evaluate turn-taking and barge-in performance.

---

[1]https://github.com/NVIDIA-NeMo/NeMo/tree/main/nemo/collections/speechlm2

Table 4: Performance of different methods on Llama Questions and Web Questions benchmark in accuracy (%). Results of baseline systems are taken from (Zeng et al., 2024). The results of SALM-Duplex are reproduced by ourselves. We use "$thk$" to denote the proposed chronological thinking.

| Method | Modality | # Params | Full-duplex | Llama Questions | Web Questions |
|---|---|---|---|---|---|
| TWIST | S→S | 7B | ✗ | 4.0 | 1.5 |
| SpeechGPT | S→T | 7B | ✗ | 21.6 | 6.5 |
| Spectron | S→T | 1B | ✗ | 21.9 | 6.1 |
| Moshi | S→S | 7B | ✓ | 21.0 | 9.2 |
| GLM-4-Voice | S→S | 9B | ✗ | 50.7 | 15.9 |
| SALM-Duplex* | S→S | 1.5B | ✓ | 15.0 | 6.7 |
| CT-Duplex w/o $thk$ | S→S | 1.7B | ✓ | 30.4 | 13.2 |
| CT-Duplex w/ $thk$ | S→S | 1.7B | ✓ | **31.4** | **13.3** |

## 4.4 RESULTS

### 4.4.1 REASONING QUALITY

We evaluate the performance of the proposed CT-Duplex model with and without the chronological thinking mechanism, denoted as CT-Duplex w/ $thk$ and CT-Duplex w/o $thk$, respectively. For comparison, we also include the method from (Hu et al., 2025), named SALM-Duplex. The network architecture of SALM-Duplex is nearly identical to ours, with the key difference being that its LLM backbone simultaneously predicts both text tokens and audio tokens, whereas our LLM backbone only predicts text tokens, and an additional Transformer decoder is used to predict audio tokens. Furthermore, by feeding the LLM backbone with the ground-truth text of user inquiries and using the generated text to calculate scores, we establish an optimal cascaded system (GT-LM) to compare with our proposed method. The evaluation result of the reasoning abilities of different methods is shown in Table 3. It can be observed that the integration of chronological thinking has enhanced response quality, **especially in scenarios requiring complex reasoning**, as evidenced by 8.75% improvements on the SpokenWOZ benchmark. For everyday multi-turn dialogues evaluated on MtBenchEval, the observed gains are relatively modest (2.09%).

Meanwhile, Table 3 shows that the GT-LM method, which uses ground-truth user inquiry texts, performs worse than the proposed chronological thinking method on SpokenWOZ, but better than the method without thinking. However, on MtBenchEval, it achieves the best performance. This is because the output obtained by GT-LM is essentially the result of the text LLM without CoT. Therefore, for scenarios that require certain reasoning, this method performs worse than the thinking-enabled CT-Duplex w/ $thk$ method. For scenarios that do not require reasoning, this method achieves optimal results due to the ideal input.

When compared to SALM-Duplex, both CT-Duplex w/ and w/o $thk$ achieve better performance. This demonstrates the effectiveness of decoupling audio token prediction from the LLM backbone.

### 4.4.2 FACTUAL KNOWLEGE CAPABILITY

We further evaluate the model's level of factual knowledge. Table 4 shows the accuracy of our proposed method compared to baseline methods on both Llama Questions and Web Questions. It can be observed that when it comes to benckmarks requiring factual knowledge, the proposed choronological thinking method showed negligible improvement. This is because factual knowledge-based Question-Answering tasks require minimal reasoning, as the model only needs to possess the relevant knowledge to answer questions correctly. Compared to SALM-Duplex, our models still achieve higher accuracy as we alleviate the predictive burden on the LLM. Meanwhile, when compared with other baseline methods, it can be observed that except for GLM-4-Voice, our models outperform all baseline approaches, even though baseline methods such as Moshi have significantly more parameters than our models (7B vs. 1.7B). Although the GLM-4-Voice method achieves much higher accuracy than our proposed models, its 9B-parameter count far exceeds that of our models. Additionally, its half-duplex structure, which is not constrained by real-time and causal requirements, also contributes to its higher accuracy. In summary, the full-duplex SDLM proposed in this paper demonstrates strong factual knowledge retention capabilities.

Table 5: Evaluation of conversational behaviors of different methods on the *Impatient* dataset proposed in (Chen et al., b), in terms of turn-taking latency, barge-in latency, and barge-in success rate. Results of baseline systems are taken from (Chen et al., b). The results of SALM-Duplex are reproduced by ourselves. We use "*thk*" to denote the proposed chronological thinking

| Method | E2E | Turn-taking | Barge-in | |
| | | Latency (↓) | Latency (↓) | Success rate (↑) |
|---|---|---|---|---|
| Freeze-Omni | ✗ | 1.17 | 1.20 | 79.50% |
| dGSLM | ✓ | 0.57 | 0.86 | 85.00% |
| Moshi | ✓ | n.a. | 0.81 | 55.10% |
| ORISE | ✓ | 0.43 | 0.61 | 96.80% |
| SALM-Duplex* | ✓ | 0.92 | 0.69 | 87.50% |
| CT-Duplex w/o *thk* | ✓ | **0.45** | **0.53** | 88.63% |
| CT-Duplex w/ *thk* | ✓ | 0.68 | 0.54 | **94.05%** |

### 4.4.3 TURN-TAKING AND BARGE-IN EVALUATION

Table 5 presents a comparison of the turn-taking and barge-in performance between the proposed method and the baseline method. It can be observed that the results are comparable with and without the thinking mechanism., whether in terms of turn-taking latency or barge-in behavior. Although the CT-Duplex w/o *thk* method achieves a lower turn (0.20s lower), this difference has a negligible impact on the user experience in dialogue systems. Furthermore, the CT-Duplex w/ *thk* method achieves a higher barge-in success rate. These results demonstrate that chronological thinking introduced in this work does not impair the full-duplex SDLM's turn-taking or barge-in abilities. That is because the proposed method generates nodes of the thinking chain chronologically with the input, introducing no additional computational overhead or extra latency. Besides, we only replace the silence tokens in the original full-duplex SDLM with thinking chain tokens, ensuring that the thinking process occurs exclusively during the listening phase without altering any response tokens of the SDLM.

### 4.4.4 SUBJECTIVE RESULTS

We conduct an A/B test to validate the subjective evaluation results of the proposed chronological thinking method, aiming to assess its performance in terms of audio fidelity and response content quality(Brachmanski). The experimental setups are detailed in Appendix A.4. The test dataset is sourced from SpokenWOZ. We select 10 fluent English speakers to evaluate the audio quality and con-

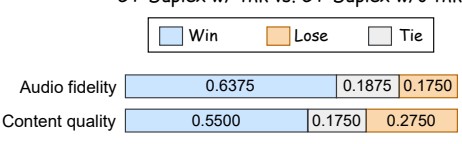

Figure 3: The A/B test results, including evaluations of both audio generation quality and response content quality.

tent of the responses generated by CT-Duplex w/ *thk* and CT-Duplex w/o *thk*. The experimental results are shown in Figure 3. It can be observed that the subjective metrics for both audio quality and response content of CT-Duplex w/ *thk* are superior to those of CT-Duplex w/o *thk*. Although the proposed thinking method is designed to improve response content quality, the enhancement in response quality leads to a lower loss in agent text prediction. This, in turn, allows the model to learn more from the audio prediction loss, thereby achieving a higher level of audio quality.

## 5 CONCLUSION

This paper proposes a chronological thinking mechanism that enables full-duplex SDLMs to possess a human-like thinking-while-listening ability during conversations. Inspired by research on human cognitive architecture, we introduce a chronological thinking chain comprising five distinct node types, each corresponding to components of the ACT-R framework. By replacing silence tokens in conventional full-duplex SDLMs with chronological thinking chain tokens, we achieve causal and no-additional-latency thinking during listening phases. Objective and subjective evaluation results demonstrate that the proposed method achieves higher response quality, especially in scenarios requiring reasoning, without compromising the turn-taking and barge-in performance of SDLMs.

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

## A APPENDIX

### A.1 DISCUSSION ABOUT STREAMING ASR AND CHRONOLOGICAL THINKING

Using streaming ASR results as thinking content is an intuitive implementation approach to enable SDLMs with the thinking-while-listening ability (Moritz et al., 2020; He et al., 2019; Yamamoto et al., 2025). Under the task scenarios and experimental setup of this paper, the performance upper bound of this streaming ASR method corresponds to the performance of the text LLM when fed with the ground-truth user inquiry text. This output reflects the LLM's response without any thinking process; when the task requires reasoning, its performance is significantly affected. In contrast, the chronological thinking method proposed in this paper, which includes analysis of user intent, contextual entity binding, logical reasoning, prerequisite knowledge extraction, and prediction of agent actions, achieves superior reasoning performance. In fact, the comparison between the GT-LM results and our method in Table 3 demonstrates this conclusion. Additionally, using streaming ASR results as thinking content is also affected by ASR accuracy. In experiments, we observe that streaming ASR models often insert padding tokens between adjacent text tokens, which disrupts semantic consistency and degrades SDLM's performance.

---

Round1:
User:
Help me order a restaurant to celebrate my birthday this weekend.
Agent:
###
[Help me]{INTENT} Request assistance
[order a restaurant]{ACTION} Initiate booking process
[to celebrate my birthday]{LOGIC} Purpose: birthday celebration
[this weekend]{ENTITY} Timeframe: weekend
{KNOWLEDGE} Birthday: Decorations, discounts, or special perks
###
Certainly! For a birthday celebration, I recommend a place that offers a special dining experience. Do you have a preferred cuisine or location in mind?

---

Round2:
User:
I'd like to try Italian food, and the place should be in the city center.
Agent:
###
[I'd like to try]{INTENT} Express cuisine
[Italian food]{ENTITY} Cuisine:Italian
[and the place should be in the city center]{LOGIC} Location:city center
###
Great choice! How about Trattoria Bella, located right in the heart of the city? It's known for its authentic Italian dishes and cozy atmosphere.

---

Round3:
User:
That sounds perfect. Can you book a table for me and three of my friends?
Agent:
###
[That sounds perfect]{INTENT} Confirmation
[Can you book a table]{ACTION} Book table
[for me]{ENTITY} 1 person
[and three of my friends?]{ENTITY} 3 people
{LOGIC} 4 people
{ENTITY} Restaurant@ctx:Trattoria Bella
###
Sure, I can book a table for four people at Trattoria Bella. What day this weekend would you like to visit?

---

Figure 4: A dialogue example with chronological thinking chains.

## A.2 Example of chronological thinking chains

Figure 4 displays a three-turn dialogue example incorporating chronological thinking chains. It includes simple semantic reasoning (inferring the preferred restaurant for a birthday celebration) and mathematical reasoning (me + 3 people = 4 people), as well as cross-turn entity tracking (Trattoria Bella). In the design of the thinking chain, we use "@ctx:ID" to bind and track entities across dialogue turns, with this functionality implemented within the ENTITY nodes. The usage of "@ctx:ID" is demonstrated in the last line of the thinking chain in the third dialogue turn.

## A.3 Completeness of Thinking Chains

Although in Section 3.3 we employ truncation to ensure that thinking tokens do not occupy the original response tokens, the proposed method significantly shortens the length of the generated thinking content by replacing natural language with structured thinking chain nodes. This maximizes the completeness of the thinking process within a given time duration. To verify the completeness of the generated thinking chains during training process, we statistically analyze the frame count of each user utterance and the corresponding token count of the thinking chains, across three training datasets. We then calcu-

Table 6: The ratio of dialogue turns where the thinking chain tokens being less than the frame count of user utterances.

| Dataset | Ratio |
| --- | --- |
| GenConv | 98.91% |
| SpokenWOZ-G | 96.78% |
| Llamaq-G | 94.69% |

late the ratio of dialogue turns where the thinking chain tokens being less than the frame count of user utterances, as summarized in Table 6. The results demonstrate that for the majority of cases across all three datasets, the token count of the thinking chains remains lower than the frame count of user utterances, indicating that most thinking chains are fully preserved.

## A.4 SUBJECTIVE EXPERIMENTS

To ensure a fair comparison, we conduct a blind A/B test where human evaluators are presented with paired responses from the full-duplex SDLM with and without chronological thinking in a random order. Evaluators are asked to select the preferred response based on audio fidelity and response content quality, or mark them as a tie if no significant difference is observed.

We recruit 10 fluent English speakers as evaluators, where each participant assesses 20 audio samples selected randomly from the test dataset. These subjects are either from English-speaking countries or possess over seven years of speaking experience, ensuring high proficiency. To ensure precise evaluation, evaluators are allowed to replay the audio repeatedly but are required to listen to each sample at least three times before providing a rating.

## A.5 LLM USAGE STATEMENT

We used ChatGPT only for minor language editing to improve clarity and conciseness. No part of theresearch idea, methodology, or analysis was generated by LLMs.

