# OpenReview forum: "Chronological Thinking in Full-Duplex Spoken Dialogue Language Models"
_ICLR.cc/2026/Conference — Submitted to ICLR 2026_

### Official Review · Reviewer_hQuT · 2025-10-31

**Soundness:** 1
**Presentation:** 3
**Contribution:** 2
**Rating:** 2
**Confidence:** 3

**Summary:**

This paper introduces a chronological thinking mechanism, incorporated within the full-duplex pipeline of spoken dialogue systems. The core motivation is to prevent the agent idle when listening to user's utterances, which is not aligned with cognitive human interactions. There are five nodes consisting of the chronological thinking, and they are implemented on top of SALM-duplex model. Experiments are based on multi-turn benchmarks (SpokenWOZ, MtBenchEval) and factual QA benchmarks (Llama Questions, Web Questions).

**Strengths:**

* The motivation is strong. Agent's idle state generation is a critical issue regarding performance and efficiency in duplex systems. This concurrent speech/reasoning processing is aligns with recent trends in this field.

* The overall writing flow is easy to read.

**Weaknesses:**

1. **Evaluation metric**:

    1. It is widely acknowledged that the evaluation performance of LLMs is highly sensitive to the choice of input prompt. However, the manuscript does not provide sufficient details regarding the prompts or the specific evaluation features used in experiments.

    1. For multi-turn dialogues, controlling conversation flow is inherently challenging. The manuscript does not clarify whether the evaluation is conducted at the turn level or at the dialogue level. For example, is the model provided with the entire dialogue history and evaluated solely on its final response?

    1. Using syntactic or semantic similarity metrics such as BLEU and Sentence-BERT is not well-suited for task-oriented dialogue systems, as these metrics do not assess task completion or goal success. Metrics that measure task success rates or completion performance would be more appropriate for this evaluation scenario.

1. **Analysis on the proposed method**: Although the authors emphasize the motivation rooted in the ACT-R cognitive framework, the manuscript lacks further analysis or ablation studies to support the claim. For instance, the five node types described in Table 1 rely entirely on LLM-generated content, yet no analysis is provided to verify whether the generated content aligns with the authors' intended definitions or how each node type contributes to performance across different conditions.

1. **Actual performance benefit of the proposed thinking mechanism**: The performance improvements between the variants with and without the proposed thinking mechanism are marginal. In particular, in Table 4, the claim that chronological thinking outperforms SALM-Duplex appears overstated. The reported gains may instead stem from (1) predicting audio tokens only and (2) the addition of a Transformer decoder module, rather than from the thinking mechanism itself.

1. **# of parameters in Table 1**: Although the proposed architecture introduces an additional Transformer decoder relative to SALM-Duplex, Table 1 does not clearly report the total number of parameters. The manuscript should provide explicit details regarding the parameter count and the architectural additions.

1. **Concerns on the figure**: Figure 2 appears visually similar to figures used in the SALM-Duplex paper, which may raise concerns about originality. It is recommended that the figure highlight this paper's distinctive contribution - specifically, the chronological thinking mechanism.

1. **Subjective results**: Information on evaluator recruitment, sample size, and evaluation guidelines is insufficient. These details are necessary to ensure reproducibility and to support the credibility of the subjective evaluation results.

1. **Some editorial suggeestions**:
    1. Terms "SpokenWOZ" after the line 312 need to be described as "SpokenWOZ-G" to prevent confusion?
    1. MtBenchEval in line 356 should be properly cited.
    1. Typo: "Impatiet" in Table 5.

**Questions:**

See weaknesses.

---

> ### Author Response · Authors · 2025-11-19
> **Response to reviewer hQuT**
>
> 1. Evaluation metric:
>     1. It is widely acknowledged that the evaluation performance of LLMs is highly sensitive to the choice of input prompt. However, the manuscript does not provide sufficient details regarding the prompts or the specific evaluation features used in experiments.
>
>     We are grateful for this suggestion and include the updated explanation in the paper. We ensure consistency with published benchmarks by utilizing the evaluation methodology and prompts from the open-sourced URO-Bench [1].
>
>     2.  For multi-turn dialogues, controlling conversation flow is inherently challenging. The manuscript does not clarify whether the evaluation is conducted at the turn level or at the dialogue level. For example, is the model provided with the entire dialogue history and evaluated solely on its final response?
>
>     For multi-turn dialogues, the response in every turn is evaluated, rather than just the response in the final turn. Full-duplex SDLMs are real-time dialogue systems rather than turn-based systems; the model continuously receives the user input stream, listening while the user speaks and responding when the user finishes speaking. Therefore, the method of feeding the dialogue history as a prompt to the LLM to evaluate the output of the final turn is not applicable. We use the multi-turn dialogue evaluation prompt from URO-bench to calculate the GPT-score of the entire conversation. This approach takes the complete set of questions and responses from the entire dialogue as input and prompts GPT to calculate the GPT-score.
>
>     3. Using syntactic or semantic similarity metrics such as BLEU and Sentence-BERT is not well-suited for task-oriented dialogue systems, as these metrics do not assess task completion or goal success. Metrics that measure task success rates or completion performance would be more appropriate for this evaluation
>
>     The primary evaluation metric, GPT-score, assesses both task success rate and response completeness. BLEU and Sentence-BERT are only provided as a reference. Experimental results indicate that these metrics demonstrate a positive correlation with GPT-score. Furthermore, we remain open to testing any additional metrics you suggest, if time permits.
>
> 2. Analysis on the proposed method: Although the authors emphasize the motivation rooted in the ACT-R cognitive framework, the manuscript lacks further analysis or ablation studies to support the claim. For instance, the five node types described in Table 1 rely entirely on LLM-generated content, yet no analysis is provided to verify whether the generated content aligns with the authors' intended definitions or how each node type contributes to performance across different conditions.
>
> We appreciate this suggestion. Actually, we have conducted ablation studies and find that all five nodes are critical to the model's overall effectiveness. The removal of specific nodes has different impacts on different test datasets. For instance, the removal of INTENT and LOGIC nodes significantly affects SpokenWOZ, while removing the KNOWLEDGE node has a more impact on MtBenchEval. This variation highlights the distinct role of each node depending on the task. Furthermore, we base the selection of these five nodes on the theory of human cognitive architecture. This theory indicates that these components together constitute a complete cognitive and reasoning process, which provides a strong theoretical foundation for our design. Additionally, as mentioned in the paper, we also explore alternative thinking mechanisms, such as using streaming ASR as the reasoning content, but observe that the results are not good.

---

> > ### Author Response · Authors · 2025-11-19
> > **Response to reviewer hQuT**
> >
> > 3. Actual performance benefit of the proposed thinking mechanism: The performance improvements between the variants with and without the proposed thinking mechanism are marginal. In particular, in Table 4, the claim that chronological thinking outperforms SALM-Duplex appears overstated. The reported gains may instead stem from (1) predicting audio tokens only and (2) the addition of a Transformer decoder module, rather than from the thinking mechanism itself.
> >
> > We appreciate your comment. We discuss this in Section 4.4.2, noting that this is because factual knowledge-based Question-Answering tasks require minimal reasoning, as the model only needs to possess the relevant knowledge to answer questions correctly (in line 417 in the original submission). In Section 4.4.2, we do not emphasize the improvement brought by chronological thinking, but demonstrate the effectiveness of our model on QA tasks. We update the description in the revised version of the paper to enhance clarity.
> >
> > We point out in the paper that our proposed thinking mechanism demonstrates more significant improvement on tasks that require reasoning, such as SpokenWOZ, while the improvement on QA tasks that do not require reasoning is more limited. This aligns with our understanding of LLM reasoning mechanisms, as CoT does not lead to improvements in all tasks.
> >
> > On the other hand, our method does not result in performance degradation on tasks that do not require reasoning, or otherwise yields marginal improvements. This demonstrates the effectiveness of our method across all tasks. Performance remains unaffected when the task does not require reasoning, whereas it improves significantly when reasoning is required.
> >
> > 4. of parameters in Table 1: Although the proposed architecture introduces an additional Transformer decoder relative to SALM-Duplex, Table 1 does not clearly report the total number of parameters. The manuscript should provide explicit details regarding the parameter count and the architectural additions.
> >
> > Thank you for your suggestion. We revise this in Table 4 in the revised paper.
> >
> > 5. Concerns on the figure: Figure 2 appears visually similar to figures used in the SALM-Duplex paper, which may raise concerns about originality. It is recommended that the figure highlight this paper's distinctive contribution - specifically, the chronological thinking mechanism.
> >
> > We are grateful for this suggestion. Our method differs from SALM-Duplex in two key aspects. First, SALM-Duplex remains idle while the user speaks, as illustrated in Figure 1(b), whereas our method engages in chronological thinking during this period, corresponding to Figure 1(c). We highlight this improvement using blue think tokens in Figure 2. Second, SALM-Duplex uses the LLM backbone to predict text tokens and audio tokens simultaneously, whereas in our method, the LLM backbone predicts only text tokens, and we use an additional Transformer decoder to predict audio tokens. This architectural distinction is also depicted in Figure 2.
> >
> > Following your suggestion, we add a red box to the Transformer decoder to highlight this component and further clarify the thinking mechanism in the figure caption. However, to preserve clarity regarding the full-duplex mechanism, we retain the primary structure of Figure 2.
> >
> > 6. Subjective results: Information on evaluator recruitment, sample size, and evaluation guidelines is insufficient. These details are necessary to ensure reproducibility and to support the credibility of the subjective evaluation results.
> >
> > Thank you for your comments. We provide a supplementary explanation of the subjective experiment settings in Appendix A.4.
> >
> > 7. Some editorial suggestions: Terms "SpokenWOZ" after the line 312 need to be described as "SpokenWOZ-G" to prevent confusion? MtBenchEval in line 356 should be properly cited. Typo: "Impatiet" in Table 5.
> >
> > Thank you for your suggestion. We correct these errors in the revised paper. We respectfully note that SpokenWOZ-G refers to our generated dataset. The text following line 312 in the original paper describes the SpokenWOZ dataset itself. The phrase "SpokenWOZ’s dialogues and generated dialogues" in line 315 refers to the original dataset and the generated dataset correspondingly; therefore, the description here is correct.
> >
> > Reference
> >
> > [1] https://github.com/Ruiqi-Yan/URO-Bench/blob/main/metrics/mark_gpt.py

---

> ### Author Response · Authors · 2025-11-27
>
> We just wanted to follow up to see if our response has sufficiently addressed your concerns. We remain available to answer any further questions or provide additional clarifications before the discussion period ends.

---

### Official Review · Reviewer_gefN · 2025-11-01

**Soundness:** 3
**Presentation:** 3
**Contribution:** 3
**Rating:** 4
**Confidence:** 5

**Summary:**

This manuscript explores the integration of deliberative reasoning capabilities into end-to-end spoken dialogue models, representing an interesting and nascent direction in the audio domain. The core technique involves a "Think-While-Listen" paradigm: specific output tokens emitted by the Language Model during the user's speaking time are substituted with corresponding textual reasoning tokens. This concurrent approach is claimed to introduce zero latency for the reasoning process. The experimental section includes basic conversational QA tests and latency measurements, reporting decent performance gains over weak, foundational baselines.

**Strengths:**

1. The work is one of the first attempts to systematically explore and implement a deliberative (or "thinking") mode within the architecture of end-to-end spoken dialogue models.

2. The proposed method, while structurally simple, is conceptually sound. Leveraging the user's silence or speaking time for concurrent processing is a valid and pragmatic approach to integrating complex reasoning without incurring additional latency.

**Weaknesses:**

1. The exploration of the reasoning mechanism is overly simplistic, relying merely on substituting special LLM output tokens with scratchpad text. Further architectural and procedural investigation is warranted. Potential avenues for future exploration include: Expanding beyond Think-While-Listen to Think-While-Speak (concurrent reasoning during both input and output phases). Introducing a dedicated reasoning output head, separate from the main dialogue response head, purely for inference and planning. Investigating whether fine-tuning the reasoning component using Reinforcement Learning (RL) post-pre-training could enhance the quality of the deliberation. Analyzing the reasoning capability's sensitivity to diverse training data distributions.
2. Many conversational scenarios do not necessitate complex reasoning. The training data generation phase should be critically optimized to focus on high-level logical inference dialogues (e.g., those found in benchmarks like BigBenchAudio in GPT-realtime). Furthermore, the examples provided in Appendix A1 appear heavily biased toward tool-use/Agent functionality (which often involves a search path). It is crucial to include more generalized conversational scenarios and provide a clearer manifestation of the ACT-R theory (or similar cognitive architectures) in these common dialogue contexts.
3. The current experimental results are limited by the selection of relatively weak baselines (e.g., GLM-4-Voice). The evaluation must be expanded to include comparisons against state-of-the-art competitive models such as Qwen3-Omni and Kimi-Audio to properly benchmark the proposed technique's efficacy.
4. The thinking paradigm is not inherently constrained to the spoken dialogue model architecture. Validation should be extended to a wider array of models and tasks, such as non-interruptive Spoken QA, and training should ideally be conducted on larger language models to confirm scalability.
5. The ablation study (Table 4) shows relatively marginal gains, suggesting that the current data and scenario design may not sufficiently highlight the benefits of the reasoning process. Further optimization of the data and evaluation scenarios is highly recommended, potentially utilizing broader benchmarks like VoiceBench.
6. The authors should include a more thorough discussion and comparison with concurrent related works [1] and [2].

[1]. Can SpeechLLMs Think while Listening?

[2]. STITCH: Simultaneous Thinking and Talking with Chunked Reasoning for Spoken Language Models

**Questions:**

N/A

---

> ### Author Response · Authors · 2025-11-19
> **Response to reviewer gefN**
>
> We respectfully wish to clarify a fundamental distinction regarding the scope of our work as we identify there may be a misunderstanding here. Our research focuses on full-duplex LLMs, whereas the models discussed in your review function primarily as turn-based LLMs. A full-duplex LLM employs a dual-stream architecture where one stream continuously receives user input while the other continuously generates output, maintaining an "Always-on" state. Within the output stream, the model autonomously determines when to initiate and cease speaking, as illustrated in Figure 1(b). This represents a fundamentally different architecture compared to turn-based LLMs. As shown in Figure 1(a), turn-based systems typically wait for the user to explicitly cease input before responding and lack the capability to handle interruptions during generation. Turn-based LLMs are incapable of processing user input in real-time, which diverges from natural human communication patterns. As stated in the introduction, this "always-on" full-duplex LLM delivers more natural, fluid, and human-like conversations, with the ability to proactively take turns, offer backchannel responses, make timely corrections, and gracefully yield when the user barges in.
>
> Regarding the recommended works in your review, such as Qwen3-omni, "Can Speech LLMs Think while Listening?", and STITCH, we note that these are turn-based LLMs. Furthermore, we respectfully point out that these are concurrent works to ours, with some appearing after the ICLR submission deadline; thus, they fall outside the scope of our analysis.
>
> Regarding "Can Speech LLMs Think while Listening?", although it is not a full-duplex LLM, its aim to eliminate latency aligns with our motivation and validates the research direction of our proposed method.
>
> About weakness:
>
> 1. The exploration of the reasoning mechanism is overly simplistic, relying merely on substituting special LLM output tokens with scratchpad text. Further architectural and procedural investigation is warranted. Potential avenues for future exploration include: Expanding beyond Think-While-Listen to Think-While-Speak (concurrent reasoning during both input and output phases). Introducing a dedicated reasoning output head, separate from the main dialogue response head, purely for inference and planning. Investigating whether fine-tuning the reasoning component using Reinforcement Learning (RL) post-pre-training could enhance the quality of the deliberation. Analyzing the reasoning capability's sensitivity to diverse training data distributions.
>
> We appreciate this insightful suggestion. The primary objective of this paper is to address the specific limitation where existing full-duplex LLMs remain idle while the user is speaking. To this end, we introduce chronological thinking to utilize this idle state, enabling the LLM to think during the listening phase.
>
> We view your suggestion to expand beyond Think-While-Listen to Think-While-Speak as a valuable but distinct research direction. Our method achieves thinking-while-listening. Some other works also explore how to enable the LLM to think-while-speaking. This aligns with the two papers you recommended. The first paper focuses on enabling the LLM to think-while-listening, while the second focuses on enabling the LLM to think-while-speaking. We also note that they are concurrent works to our submission.
>
> This paper validates that the proposed chronological thinking method improves response quality without incurring any additional cost. Consequently, we consider methods such as thinking-while-speaking and Reinforcement Learning (RL) as natural extensions for future work to further enhance the output phase. We plan to incorporate these directions in our future research on full-duplex systems.

---

> > ### Author Response · Authors · 2025-11-19
> > **Response to reviewer gefN**
> >
> > 2. Many conversational scenarios do not necessitate complex reasoning. The training data generation phase should be critically optimized to focus on high-level logical inference dialogues (e.g., those found in benchmarks like BigBenchAudio in GPT-realtime). Furthermore, the examples provided in Appendix A1 appear heavily biased toward tool-use/Agent functionality (which often involves a search path). It is crucial to include more generalized conversational scenarios and provide a clearer manifestation of the ACT-R theory (or similar cognitive architectures) in these common dialogue contexts.
> >
> > This recommendation aligns well with our findings. The SpokenWOZ dataset used in this paper includes various reasoning tasks, such as mathematical and temporal reasoning. Experimental results demonstrate that our method achieves the most significant improvements on SpokenWOZ, while improvements on tasks requiring minimal reasoning are more marginal. This observation aligns with your perspective. On the other hand, our method does not lead to performance degradation on tasks that do not require reasoning; instead, it maintains performance or yields slight improvements. This demonstrates the robustness of our method across different tasks: performance remains stable when reasoning is unnecessary, but improves significantly when reasoning is required.
> >
> > 3. The current experimental results are limited by the selection of relatively weak baselines (e.g., GLM-4-Voice). The evaluation must be expanded to include comparisons against state-of-the-art competitive models such as Qwen3-Omni and Kimi-Audio to properly benchmark the proposed technique's efficacy.
> >
> > We respectfully note that both Qwen3-omni and Kimi-Audio operate as turn-based models. As noted above, these architectures differ significantly from the full-duplex LLMs investigated in this paper. We also observe a substantial disparity in parameter scale between these models and our approach. Furthermore, since Qwen3-omni represents concurrent work relative to our submission, we do not include it in our comparative analysis.
> >
> > 4. The thinking paradigm is not inherently constrained to the spoken dialogue model architecture. Validation should be extended to a wider array of models and tasks, such as non-interruptive Spoken QA, and training should ideally be conducted on larger language models to confirm scalability.
> >
> > We acknowledge this comment. As discussed previously, this paper focuses on full-duplex LLMs, which differ fundamentally from the majority of existing turn-based systems. We design our method specifically for full-duplex architectures, and the aspects evaluated in this paper cover those addressed in current full-duplex literature [3][4], including Spoken QA in Section 4.4.2.
> >
> > 5. The ablation study (Table 4) shows relatively marginal gains, suggesting that the current data and scenario design may not sufficiently highlight the benefits of the reasoning process. Further optimization of the data and evaluation scenarios is highly recommended, potentially utilizing broader benchmarks like VoiceBench.
> >
> > As noted in the paper, our proposed thinking mechanism demonstrates more significant improvement on tasks that require reasoning, such as SpokenWOZ, while the improvements on daily dialogue tasks and QA tasks that do not require complex reasoning are more limited. This aligns with our understanding of LLM reasoning mechanisms, as CoT does not lead to improvements in all tasks.
> >
> > On the other hand, our method does not result in performance degradation on tasks that do not require reasoning, or otherwise yields marginal improvements. This demonstrates that across all tasks, the performance of our model remains unaffected when the task does not require reasoning, whereas it improves significantly when reasoning is required.
> >
> > 6. The authors should include a more thorough discussion and comparison with concurrent related works [1] and [2].
> >
> > Thank you for your suggestion. First, we respectfully note that neither work operates as a full-duplex system; instead, they function as turn-based models. Second, we regard these papers as concurrent work. Reference [1] appears on arXiv on October 7th, which occurs after our submission deadline. Reference [2] appears on July 21st, only two months prior to our deadline. Moreover, a fundamental difference exists: [2] focuses on thinking-while-speaking, whereas our study concentrates on thinking-while-listening.

---

> > > ### Author Response · Authors · 2025-11-19
> > > **Response to reviewer gefN**
> > >
> > > Reference
> > >
> > > [1] Shih, Y. J., Raj, D., Wu, C., Zhou, W., Bong, S. K., Gaur, Y., ... & Seltzer, M. (2025). Can Speech LLMs Think while Listening?. arXiv preprint arXiv:2510.07497.
> > >
> > > [2] Chiang, C. H., Wang, X., Li, L., Lin, C. C., Lin, K., Liu, S., ... & Wang, L. (2025). Stitch: Simultaneous thinking and talking with chunked reasoning for spoken language models. arXiv preprint arXiv:2507.15375.
> > >
> > > [3] Défossez, A., Mazaré, L., Orsini, M., Royer, A., Pérez, P., Jégou, H., ... & Zeghidour, N. (2024). Moshi: a speech-text foundation model for real-time dialogue. arXiv preprint arXiv:2410.00037.
> > >
> > > [4] Yu, W., Wang, S., Yang, X., Chen, X., Tian, X., Zhang, J., ... & Zhang, C. (2024). Salmonn-omni: A codec-free llm for full-duplex speech understanding and generation. arXiv preprint arXiv:2411.18138.

---

> ### Author Response · Authors · 2025-11-27
>
> We hope our previous response effectively resolves the issues you raised. If there are any remaining uncertainties, please let us know so we can address them promptly.

---

### Official Review · Reviewer_12Gj · 2025-11-01

**Soundness:** 3
**Presentation:** 3
**Contribution:** 3
**Rating:** 6
**Confidence:** 4

**Summary:**

This paper proposes Chronological Thinking, a strictly causal, on-the-fly reasoning mechanism for full-duplex Spoken Dialogue Language Models (SDLMs) that allows them to "think while listening" instead of outputting repeated silence tokens. The system replaces silence tokens during the listening phase with structured reasoning nodes inspired by ACT-R cognitive architecture, enabling better real-time conversational reasoning without added latency. The contributions include (a) the CT-SDLM architecture integrating chronological thinking, (b) a synthetic multi-speaker dialogue dataset generation pipeline, (c) objective and subjective evaluations showing improved reasoning quality, and (d) comparisons against LLM and SDLM baselines such as SALM-Duplex and GT-LM.

**Strengths:**

1. Introduces a novel and practical causal reasoning mechanism tailored for full-duplex spoken dialogue, addressing a well-defined gap in how agents utilize the listening phase.
2. Demonstrates improvements in reasoning-heavy dialogue tasks without cost to latency or conversational dynamics.
3. Evaluation includes comparisons to baselines, across multiple metrics (GPT score, BLEU, Sentence-BERT, factual QA accuracy, turn-taking/barge-in), offering a thorough empirical picture.

**Weaknesses:**

1. Heavy reliance on synthetic datasets may limit conclusions about real-world conversational robustness and generalization.
2 The proposed improvement in factual QA performance is negligible, indicating the method’s gains may be task-specific.
3. Lack of ablation isolating the benefit of ACT-R-inspired node types versus more naive streaming reasoning approaches beyond brief mentions.
4. Architecture depends on a specific multi-component pipeline (speech encoder, LLM backbone, transformer decoder) which may reduce reproducibility or applicability for researchers without comparable resources.

**Questions:**

- How does the chronological thinking mechanism perform on purely real, noisy speech data and spontaneous interruptions, beyond synthetic datasets?
- Could you provide detailed ablation results comparing ACT-R node structuring versus unstructured incremental reasoning to confirm the design choice?

---

> ### Author Response · Authors · 2025-11-19
> **Response to reviewer 12Gj**
>
> 1. Heavy reliance on synthetic datasets may limit conclusions about real-world conversational robustness and generalization.
>
> Thank you for your comment. We wish to emphasize that our model is well-equipped to handle real human speech. Primarily, the speech encoder is pre-trained on real-world ASR data, making it naturally suited for processing real human speech. Additionally, following the initial submission, we conduct further investigations to enhance this capability. We observe that linearly mixing noise during training and employing strategies like SpecAug [1] enhance performance on real speech data. For instance, with the inclusion of noise mixing, the model attains a 100% response rate on the Voicebench CommonEval benchmark, which consists of real human speech.
>
> 2. The proposed improvement in factual QA performance is negligible, indicating the method’s gains may be task-specific.
>
> We appreciate your comment. We discuss this in Section 4.4.2, noting that this is because factual knowledge-based Question-Answering tasks require minimal reasoning, as the model only needs to possess the relevant knowledge to answer questions correctly (in line 417 in the original submission). We point out in the paper that our proposed thinking mechanism demonstrates more significant improvement on tasks that require reasoning, such as SpokenWOZ, while the improvement on QA tasks that do not require reasoning is more limited. This aligns with our understanding of LLM reasoning mechanisms, as CoT does not lead to improvements in all tasks.
>
> On the other hand, our method does not result in performance degradation on tasks that do not require reasoning, or otherwise yields marginal improvements. This demonstrates the effectiveness of our method across all tasks. Performance remains unaffected when the task does not require reasoning, whereas it improves significantly when reasoning is required.
>
> 3. Lack of ablation isolating the benefit of ACT-R-inspired node types versus more naive streaming reasoning approaches beyond brief mentions.
>
> We appreciate this insightful recommendation. We discuss the comparison with other thinking methods in Appendix A.1. As noted in the paper, we experiment with using streaming ASR as the reasoning content, but find that the results are suboptimal. Furthermore, traditional CoT is not directly applicable to the thinking-while-listening process, as it fails to guarantee causality and its length negatively impacts real-time performance. We respectfully note that, to the best of our knowledge, no other methods are available for streaming thinking-while-listening prior to the submission of this work.
>
> 4. Architecture depends on a specific multi-component pipeline (speech encoder, LLM backbone, transformer decoder) which may reduce reproducibility or applicability for researchers without comparable resources.
>
> The architecture in this paper supports a diverse range of component pipelines. For instance, regarding the LLM backbone, we utilize models such as TinyLlama-v1.1, Qwen2.5-1.5B-Instruct, and Qwen2.5-7B-Instruct. Our method is not limited to any specific speech encoder or LLM backbone; the framework remains applicable as long as the speech encoder supports streaming capabilities. Furthermore, we plan to open-source our work to provide further clarity and facilitate future research.

---

### Official Review · Reviewer_pEmW · 2025-11-02

**Soundness:** 3
**Presentation:** 3
**Contribution:** 3
**Rating:** 6
**Confidence:** 3

**Summary:**

The paper proposes Chronological Thinking (CT) for full-duplex spoken dialogue LLMs: while the user is speaking, the model fills the silence/idle slots with a causal, interruptible 5-node "thinking" chain (Entity / Intent / Action / Knowledge / Logic), so that when speech input ends, the model can answer with little extra latency. This directly targets the "inner monologue but unused" issue in Moshi and the "asynchronous thinking" idea in SALMONN-omni. On synthetic duplex dialogue sets, CT improves text/speech quality over its own duplex baseline, while keeping turn-taking roughly in the same range as prior work.

**Strengths:**

- Clear target & mechanism. Reusing silence tokens for structured, causal planning is simple and fits current full-duplex pipelines.

- Aligned with 2025 trend. Very close to SCoT, "Can Speech LLMs Think while Listening?", and SHANKS, all of which seek reasoning-while-listening in streaming setups. CT is a reasonable variant in this space.

- Empirical uplift on its own data. Within their synthetic setup, "with CT" beats "no CT", so the idea is at least self-consistent.

**Weaknesses:**

- Novelty over Moshi/SALMONN-omni/SCoT is modest. All of these already run an inner or asynchronous text stream during duplex; CT mostly adds a fixed 5-slot structure and a replacement rule. The paper should say why this is better.

- Comparisons miss the closest 2025 baselines (SCoT, SHANKS, "Can Speech LLMs Think while Listening?"), so it’s hard to attribute the gains to CT itself.

- Latency claim is fragile. Many duplex systems are at ~200–400 ms E2E; any 200–300 ms overhead from longer thinking streams is user-visible, but the paper does not give distributional or hardware-normalized numbers. (Contrast Moshi’s 160–200 ms.)

- Data realism. Most evidence is on TTS/synthetic duplex; no noisy/overlapped human speech to show the CT trigger is robust. This is exactly where recent SALMONN-omni reports strength.

**Questions:**

- Can you run CT on one public recipe (Moshi streaming eval, SALMONN-omni duplex tasks, or SCoT’s streaming CoT benchmark) to show cross-setup gains?

- Do we really need all 5 node types? Please give ablations (2-3 nodes, or a single "thinking token").

- How is CT triggered online (VAD, ASR partial, fixed frame)? What happens if the user talks unusually fast/slow?

---

> ### Author Response · Authors · 2025-11-19
> **Response to reviewer pEmW.**
>
> About Weaknesses
>
> 1. Novelty over Moshi/SALMONN-omni/SCoT is modest. All of these already run an inner or asynchronous text stream during duplex; CT mostly adds a fixed 5-slot structure and a replacement rule. The paper should say why this is better.
>
> We respectfully wish to clarify the novelty of our work relative to these models. Existing full-duplex models, such as Moshi and SALMONN-omni, typically maintain an idle state without internal reasoning while the user is speaking, which corresponds to the mechanism illustrated in Figure 1(b). Our method advances this paradigm by enabling the LLM to chronologically think during the listening process, as depicted in Figure 1(c). We believe this capability highlights the novelty of our contribution. Regarding SCoT, we respectfully note that it is a concurrent work to ours. It was submitted to arXiv in October 2025, which was after ICLR's submission deadline.
>
> 2. Comparisons miss the closest 2025 baselines (SCoT, SHANKS, "Can Speech LLMs Think while Listening?"), so it’s hard to attribute the gains to CT itself.
>
> We appreciate these valuable recommendations. However, we respectfully point out that these papers are concurrent work to our submission. The arXiv records indicate October 2nd for SCoT, October 8th for SHANKS, and October 7th for "Can Speech LLMs Think while Listening?". These dates are subsequent to the ICLR submission deadline.
>
> 3. Latency claim is fragile. Many duplex systems are at ~200–400 ms E2E; any 200–300 ms overhead from longer thinking streams is user-visible, but the paper does not give distributional or hardware-normalized numbers. (Contrast Moshi’s 160–200 ms.)
>
> Thank you for your comment. We believe that having the LLM respond immediately after the user finishes speaking, as models like Moshi do, is unnatural. You may have also noticed during your experience with Moshi that it frequently interrupts. Therefore, in the experimental setup of this paper, we intentionally introduced a 320ms (equivalent to 4 tokens) delay from the end of the user's speech to the LLM's reply, explicitly training the model to respond only after this 320ms latency. Consequently, when calculating the true response latency of our method, this 320ms offset should be subtracted. We have added clarification regarding this point in the updated version of our paper.
>
> 4. Data realism. Most evidence is on TTS/synthetic duplex; no noisy/overlapped human speech to show the CT trigger is robust. This is exactly where recent SALMONN-omni reports strength.
>
> We wish to emphasize that our model is well-equipped to handle real human speech. Primarily, the speech encoder is pre-trained on real-world ASR data, making it naturally suited for processing real human speech. Additionally, following the initial submission, we conduct further investigations to enhance this capability. We observe that linearly mixing noise during training and employing strategies like SpecAug [1] enhance performance on real speech data. For instance, with the inclusion of noise mixing, the model attains a 100% response rate on the Voicebench CommonEval benchmark, which consists of real human speech.
>
> About Questions
>
> 1. Can you run CT on one public recipe (Moshi streaming eval, SALMONN-omni duplex tasks, or SCoT’s streaming CoT benchmark) to show cross-setup gains?
>
> Thank you for your suggestion. As the training pipelines for these works are not publicly available, we are unable to apply our chronological thinking method to them for direct comparison. Regarding the evaluation benchmarks, we select SpokenWOZ and MtBenchEval because they are established public multi-turn datasets that provide a robust basis for evaluating full-duplex dialogue. The primary focus of our method is to enhance response quality in reasoning-intensive tasks, rather than optimizing for interaction metrics such as interruption or response latency. Therefore, extensive real-time evaluations are not conducted, as we believe the validation in Table 5 is sufficient.

---

> > ### Author Response · Authors · 2025-11-19
> >
> > 2. Do we really need all 5 node types? Please give ablations (2-3 nodes, or a single "thinking token")
> >
> > We appreciate this suggestion. Actually, we have conducted ablation studies and find that all five nodes are critical to the model's overall effectiveness. The removal of specific nodes has different impacts on different test datasets. For instance, the removal of INTENT and LOGIC nodes significantly affects SpokenWOZ, while removing the KNOWLEDGE node has a more impact on MtBenchEval. This variation highlights the distinct role of each node depending on the task. Furthermore, we base the selection of these five nodes on the theory of human cognitive architecture. This theory indicates that these components together constitute a complete cognitive and reasoning process, which provides a strong theoretical foundation for our design. Additionally, as mentioned in the paper, we also explore alternative thinking mechanisms, such as using streaming ASR as the reasoning content, but observe that the results are not good.
> >
> > 3. How is CT triggered online (VAD, ASR partial, fixed frame)? What happens if the user talks unusually fast/slow?
> >
> > We respectfully clarify that variations in the user's speaking rate do not impact our method. The configuration of our system remains independent of speaking speed,, a characteristic that is also consistent with existing full-duplex architectures.
> >
> > Reference
> >
> > [1] Park, D. S., Chan, W., Zhang, Y., Chiu, C. C., Zoph, B., Cubuk, E. D., & Le, Q. V. (2019). Specaugment: A simple data augmentation method for automatic speech recognition. arXiv preprint arXiv:1904.08779.

---

### Author Response · Authors · 2025-11-19
**General response to all reviewers**

We sincerely appreciate the reviewers for their dedicated efforts. We upload a revised version of this paper that incorporates several updates: 1. an explanation regarding response latency settings; 2. enhanced emphasis in Figure 2 on the improvements of our method compared to existing approaches; 3. a supplementary explanation of the subjective experiment setup; 4. error corrections and writing clarifications. We address specific concerns in detail within our individual responses to each reviewer.

---

### Meta-Review · Area_Chair_ZyPk · 2026-01-12

**Summary:**

- Concern about the novelty compared to concurrent works and existing full-duplex models. The contribution of adding a structured 5-node thinking chain might be incremental.

- A lack of comparison against the most relevant concurrent works, making it hard to attribute gains to the proposed method.

- The comparison against "relatively weak baselines" instead of state-of-the-art models.

- Concerns about the appropriateness of metrics, and lack of prompt details, and unclear evaluation protocol.

- More thorough ablations are requested.

- The latency claims "fragile," lacking distributional or hardware-normalized numbers to substantiate the "zero added latency" claim in a realistic context.

- The reported performance improvements are marginal.

- Concern about heavy reliance on synthetic/TTS data and the lack of evidence on real, noisy, or overlapped human speech.

**Reviewer Concerns:**

Addressed Concerns:

- Latency Explanation: The authors clarified their intentional 320ms response delay in training.

- Real Speech: The authors asserted their model's capability on real speech, citing pre-training on ASR data and post-submission improvements using noise mixing and SpecAug.

- Ablation Studies: The authors stated they conducted ablations, finding all five nodes critical with task-dependent impacts, and that alternative mechanisms (like streaming ASR) performed worse.

- Evaluation: The authors committed to adding prompts and clarifying multi-turn evaluation methodology.



Unaddressed Concerns:

- Novelty & Comparisons: The core issue remains. The rebuttal defends novelty based on the full-duplex focus and chronological thinking during idle time, but does not provide new experiments comparing against the conceptually closest works.

- Robustness of Gains: The explanation for marginal gains is noted, but the reviewer's concern that gains might stem from architectural changes rather than the thinking mechanism itself is not conclusively disproven.

- Comprehensive Evaluation on Real/Diverse Data: While claims of real-speech capability are made, no new results are provided in the rebuttal.

- Latency Analysis: The request for detailed, distributional, or hardware-normalized latency comparisons to strong baselines like Moshi is not addressed.

**Reviewer Scores:**

Reviewer hQuT and Reviewer gefN may raise the scores, and other Reviewers may keep the original scores.

---

### Decision · Program_Chairs · 2026-01-26

Reject